# Rare Heterozygous *PCSK1* Variants in Human Obesity: The Contribution of the p.Y181H Variant and a Literature Review

**DOI:** 10.3390/genes13101746

**Published:** 2022-09-27

**Authors:** Evelien Van Dijck, Sigri Beckers, Sara Diels, Tammy Huybrechts, An Verrijken, Kim Van Hoorenbeeck, Stijn Verhulst, Guy Massa, Luc Van Gaal, Wim Van Hul

**Affiliations:** 1Centre of Medical Genetics, University of Antwerp and Antwerp University Hospital, 2650 Edegem, Belgium; 2Department of Endocrinology, Diabetology and Metabolic Diseases, Antwerp University Hospital, 2650 Edegem, Belgium; 3Department of Pediatrics, Antwerp University Hospital, 2650 Edegem, Belgium; 4Department of Pediatrics, Jessa Hospital, 3500 Hasselt, Belgium

**Keywords:** proprotein convertase subtilisin/kexin type 1 (*PCSK1*), rare variants, founder mutation, obesity, overweight

## Abstract

Recently, it was reported that heterozygous *PCSK1* variants, causing partial PC1/3 deficiency, result in a significant increased risk for obesity. This effect was almost exclusively generated by the rare p.Y181H (rs145592525, GRCh38.p13 NM_000439.5:c.541T>C) variant, which affects PC1/3 maturation but not enzymatic capacity. As most of the identified individuals with the heterozygous p.Y181H variant were of Belgian origin, we performed a follow-up study in a population of 481 children and adolescents with obesity, and 486 lean individuals. We identified three obese (0.62%) and four lean (0.82%) p.Y181H carriers (*p* = 0.506) through sanger sequencing and high resulting melting curve analysis, indicating no association with obesity. Haplotype analysis was performed in 13 p.Y181H carriers, 20 non-carriers (10 with obesity and 10 lean), and two p.Y181H families, and showed identical haplotypes for all heterozygous carriers (*p* < 0.001). Likewise, state-of-the-art literature concerning the role of rare heterozygous *PCSK1* variants implies them to be rarely associated with monogenic obesity, as first-degree carrier relatives of patients with PC1/3 deficiency are mostly not reported to be obese. Furthermore, recent meta-analyses have only indicated a robust association for scarce disruptive heterozygous *PCSK1* variants with obesity, while clinical significance is less or sometimes lacking for most nonsynonymous variants.

## 1. Introduction

Obesity is a common disease and has recently even reached pandemic proportions, with about 650 million adults affected in 2016 [1]. The disease is characterized by an excessive amount of accumulated fat tissue, which results from an imbalance in energy consumption and expenditure. As it is mostly caused by a combination of genetic and environmental factors, it is denoted as a complex disease. Nonetheless, monogenic forms of obesity have also been identified. Particularly, single-gene pathogenic variants in genes belonging to the leptin-melanocortin signaling pathway, which is fundamental to the regulation of energy homeostasis, have been shown to independently cause obesity [2]. One of these genes is the proprotein convertase subtilisin/kexin type 1 (*PCSK1*) gene encoding proprotein convertase 1/3 (PC1/3). This protein is responsible for the proper cleavage of a number of neuropeptides and peptide hormones which play an important role in metabolism and appetite regulation. For example, pro-opiomelanocortin (POMC) is processed by PC1/3 into adrenocorticotropic hormone, which is later processed into biologically active peptides such as the anorexic α-melanocyte stimulating hormone [3,4].

The first implication of *PCSK1* in obesity arose from a patient with extreme childhood obesity described in 1995 [5]. This patient was affected by early-onset obesity and various endocrine-related disorders. She was later diagnosed with PC1/3 deficiency after discovering compound heterozygous pathogenic variants in *PCSK1* [6]. Additional clinical case investigations have since led to the discovery of 34 more cases of congenital recessive PC1/3 deficiency, corresponding to 29 distinct *PCSK1* variants (Table 1). Cases up until 2019 [7,8,9,10,11,12,13,14,15,16,17] are reviewed in a paper by Pépin et al. [7]. Additional cases have since been reported [8,9,10,11,12,13,14]. All patients carried either homozygous or compound heterozygous *PCSK1* variants and suffered from severe early malabsorptive diarrhea (within the first three months of life), which was the main clinical feature driving subsequent genetic *PCSK1* analysis. Other common traits among patients include severe early-onset obesity (with hyperphagic behavior) and several endocrine disorders, including growth hormone deficiency, abnormal glucose homeostasis, hypogonadotropic hypogonadism, hypocortisolism, elevated plasma proinsulin and pro-opiomelanocortin, and very low insulin levels [7].

The first report on the contribution of rare non-synonymous heterozygous *PCSK1* variants to obesity dates from 2012 [26]. In this study, eight novel heterozygous *PCSK1* variants were detected in a cohort of 845 non-consanguineous extremely obese Europeans. Their prevalence was next assessed by genotyping a large cohort of European individuals with obesity (*n* = 6233) and lean individuals (*n* = 6274). Three out of the eight previous rare heterozygous non-synonymous *PCSK1* variants were detected, resulting in a combined 8.7-fold higher risk of obesity. Interestingly, borderline significance (OR 7.3, *p* = 0.072) was obtained for the p.Y181H (rs145592525, GRCh38.p13 NM_000439.5:c.541T>C) variant, found in nine obese and two lean persons, which predominantly drove the overall association. Functional analysis of this variant showed impaired PC1/3 maturation with normal enzymatic activity. The two other variants, p.M125I (rs146545244, GRCh38p13 NM_000439.5:c.375G>A) and p.N180S (rs750845408, GRCh38.p13 NM_000439.5:c.539A>G), were found in one and two individuals with obesity respectively and were undetected in the lean cohort. They both displayed significantly reduced enzymatic activity, although this was cell-type specific for p.N180S [26].

The p.Y181H variant had an especially high prevalence in Belgian subjects in the previous study [26], prompting us to further evaluate the association of this variant with obesity in Belgian cohorts.

## 2. Materials and Methods

### 2.1. Study Population

Unrelated children and adolescents (adolescent if age 12–18) with overweight or obesity were recruited at the Child Obesity Clinics from the Antwerp University Hospital (Edegem, Belgium) and Jessa Hospital (Hasselt, Belgium), resulting in a total of 481 individuals (211 boys, 270 girls), of which 376 have already been described in the previous study [26]. The Flemish Growth Charts 2004 [27] were used to distinguish individuals who were overweight or obese by applying the percentile lines that cross body mass index (BMI) 25 and 30 kg/m² at 18 years of age on the Flemish age- and sex-specific BMI growth curves, respectively [28]. Four hundred and eighty-six healthy, lean, and unrelated individuals (18.5 kg/m² ≤ BMI < 25 kg/m²; 167 men, 319 women) were recruited among couples seeking prenatal counselling at the Centre for Medical Genetics in Antwerp (due to increased triple test or high maternal age) and among employees from the Antwerp University Hospital and the University of Antwerp. Population characteristics are summarized in Table 2. We also recruited family members from two Belgian p.Y181H carriers to confirm the p.Y181H haplotype. Pedigrees for both families are shown in Figure 1. The study protocol was approved by the local ethics committee. All participants gave their written informed consent, in accordance with the Declaration of Helsinki.

### 2.2. Genotyping

Genotyping was performed on genomic DNA isolated from whole blood by standard procedures [29]. We screened *PCSK1* (RefSeq NM_000439.5) exon 4 by direct sequencing to identify p.Y181H carriers among the 105 additional children and adolescents with obesity that were not already screened in the previous study. Sequencing was performed with ABI BigDye Terminator v1.1 Cycle Sequencing kits (Thermo Fisher Scientific, Waltham, MA, USA) on an ABI Prism Genetic Analyzer 3130xl (Applied Biosystems Inc., Foster City, CA, USA) at the Centre of Medical Genetics, Antwerp.

Screening of exon 4 in the lean control population was performed by high-resolution melting curve (HRM) analysis on a Lightcycler LC480 Real-Time PCR System (Roche, Penzberg, Germany) at the Centre of Medical Genetics, Antwerp. This technique was validated to detect the p.Y181H variant. Real-time PCR was performed in a reaction volume of 10 µL. The amplification mixture included 10 ng of template DNA, 1.5 mM MgCl2, 0.15 mM dNTP’s, 500 nM primers, 0.015 U/µL GoTaq (Promega Corporation, Madison, WI, USA), 1× GoTaq buffer, and the saturating dye LCGreen+ (Idaho Technology, Salt Lake City, UT, USA) in a 0.5× concentration. Following amplification, samples were denaturated at 95 °C, renaturated at 40 °C, and then melted between 60 °C and 90 °C while constantly measuring fluorescence. Samples with resulting melting curves deviating from wild type were additionally confirmed by Sanger sequencing, as described above. Primer sequences for HRM and sequencing are available upon request.

### 2.3. Haplotype Analysis

To determine the haplotype on which the p.Y181H variant is present, we sequenced intron 3 and 4 for all Belgian p.Y181H variant carriers (*n* = 13), 20 non-carriers (10 with obesity and 10 lean subjects, randomly selected from the populations) and the two Belgian p.Y181H families by the methods described above. We identified four polymorphisms in intron 3 for haplotype analysis: rs155994 (NM_000439.5:c.396+536A>T), rs148508600 (NM_000439.5:c.396+772C>T), rs459608 (NM_000439.5:c.397-449T>G), and rs456709 (NM_000439.5:c.397-260G>A), and estimated haplotypes for the selected individuals with the FamHap program [30,31].

### 2.4. Statistical Analysis

The p.Y181H carrier frequency was compared by Fisher’s exact test. A power calculation was conducted with the online Genetic Association Study Power Calculator [32]. With our study size, we have adequate power (>80%, *p* = 0.05) to detect a genotype relative risk of 3.0 with a minor allele frequency (MAF) of 0.6% for all disease models in our cases versus controls setup. The above value estimates are within the bounds reported by Creemers et al. [26], with disease allele frequency based on the p.Y181H prevalence in the Belgian control population and a relative risk based on the prevalence of the p.Y181H variant in the cases vs controls. The haplotype frequency between *PCSK1* p.Y181H carriers and non-carriers was compared by chi-square analysis. All statistical analyses were performed using SPSS version 26.0 (SPSS, Chicago, IL, USA). The significance level was set at *p* = 0.05. No additional analyses concerning bias or population stratification were conducted, as both cohorts are from unrelated Belgian origins.

### 2.5. Literature Search

An extensive literature search was conducted (March 2022) on PubMed using specific search terms in the title and/or abstract of peer-reviewed publications: *PCSK1* AND one or a combination of the following: genetic variant, mutation, obesity, weight, and BMI. Only publications in the English language were considered from 1997 onwards, as this date marks the first time *PCSK1* was associated with obesity [6].

## 3. Results

After the initial observations by Creemers et al. [26]—that the *PCSK1* p.Y181H variant was more prevalent among individuals with obesity and was especially prevalent among Belgian individuals—we decided to further evaluate this variant in a Belgian cohort. A total of 481 children and adolescents with overweight or obesity and a group of 486 healthy, lean subjects were included. We identified three patients with obesity (0.62 %) and four lean (0.82%) controls carrying the heterozygous p.Y181H variant (*p* = 0.506).

Since we (i) did not find a statistically significant difference in p.Y181H carrier frequency between patients with overweight or obesity and lean individuals and (ii), our average carrier frequency was elevated compared to the European Non-Finnish MAF reported in gnomAD [33] (0.72% vs. 0.021%, respectively, *p* < 0.001), and we hypothesized that p.Y181H could possibly be a founder variant in this population. To further investigate this hypothesis, the haplotype for four polymorphisms (rs155994, rs442627, rs459608, and rs456709, respectively) in the neighboring region for 13 Belgian p.Y181H carriers and 20 non-carriers was generated. A GTGA haplotype was shared by all p.Y181H carriers and only four non-carriers (20%) had the same genetic background (*p* < 0.001). Furthermore, this haplotype was determined in two p.Y181H families and segregation of the p.Y181H variant with the GTGA haplotype was confirmed, whilst this haplotype was absent in all non-carriers in these families (Figure 1).

From the literature search, a total of 21 hits were found describing cases of *PC1/3* deficiency, which are summarized in Table 1. Seven additional publications, including rare heterozygous *PCSK1* variants in the context of obesity, are incorporated in the discussion section.

## 4. Discussion

The association of obesity with rare homozygous or compound heterozygous *PCSK1* variants is well established through PC1/3 deficiency and goes back as far as 1997 [6]. More recently, a study to which we contributed implied that rare heterozygous *PCSK1* variants also significantly contribute to human obesity [26]. In this study, three novel rare heterozygous non-synonymous *PCSK1* variants (p.Y181H, p.N180S, and p.M125I), which lead to partial PC1/3 deficiency, were found to jointly increase the risk of obesity by 8.7-fold. Importantly, the p.Y181H variant was the main driver as an individual OR of 7.3 was found at borderline significance (*p* = 0.072), while the other variants (p.N180S and p.M125I) had a smaller contribution as they were less frequent (identified in two and one patients with obesity, respectively, OR and p-values not separately mentioned). The p.Y181H variant was found in nine individuals with obesity, making it the most prevalent variant in the population with obesity. Of the nine carriers with obesity identified, five were Belgian (0.17% of 2872 Belgian cases with obesity), three were Swiss (0.18% of 1662 Swiss cases with obesity), and one was French (0.07% of 1526 French cases with obesity). Remarkably, the p.Y181H variant was the only variant for which lean carriers were identified: one in a Belgian adult (0.65% of 154 Belgian lean controls) and one in a French (0.07% of 1406 French lean controls) adult. The high prevalence of the p.Y181H variant in both Belgian cohorts (with obesity and lean) initiated our follow-up study, in which we conversely identified equal frequencies of the p.Y181H variant in our patients with obesity and healthy weight individuals, implying no association between p.Y181H and obesity for our Belgian sampled cohort. Additionally, the possibly pathogenic p.N180S variant, which was found in two unrelated patients with obesity and not in the lean cohort by Creemers et al. [26], has now been reported in a lean adult (BMI = 20.2) and his lean son (BMI = 21.5) [34].

Together, these findings currently question the substantial involvement of rare heterozygous *PCSK1* variants in obesity, as described by Creemers et al. [26]. We suspect that previous results might have been influenced by population stratification due to the unequal distribution of ethnicities in the population with obesity (46% Belgian, 27% Swiss, 24% French, 3% Finnish) compared to the lean population (75% Finnish, 22.5% French, 2.5% Belgian), despite efforts to exclude this type of confounding by correcting for geographical area. There are indeed clear ethnic discrepancies in *PCSK1* p.Y181H MAF in gnomAD [33], even within the European populations. The North-Western European population has the highest MAF (0.036%), followed by the Other non-Finnish European population (0.016%). The MAF is clearly lower for the Swedish (0.008%) and Southern-European (0.009%) populations, and is even completely lacking from the Finnish population, as well as from the Bulgarian and Estonian populations, with the remark that these last two have quite restricted cohort sizes. Along these lines, the p.Y181H variant was not found in a cohort of 812 Norwegian individuals (485 with morbid obesity and 327 normal weight) screened for *PSCK1* variants [35]. A study by Kleinendorst et al. [36], however, could identify two patients from Dutch centers with obesity who carry the p.Y181H mutation from a patient cohort of 1230 individuals (MAF = 0.08%). The above reported MAFs of the p.Y181H variant clearly indicate a higher prevalence in several Central European populations (Belgian, Swiss, Dutch) compared to Northern (Sweden, Norway, Finland, Estonia) and Southern European populations. As the MAF is especially high in the Belgian population, we suggest the possibility of an ancient founder mutation, as was implied by our haplotype analysis.

Regarding the possible involvement of (other) rare heterozygous *PCSK1* variants in obesity, to date, only one heterozygous *PCSK1* variant (p.Arg80*, Grch38.p13 NM_000439.5:c.238C>T) was reported to co-segregate with obesity [34]. A new case report by Qian et al. [12], however, refutes these findings. Their patient carried the homozygous p.Arg80* variant and was diagnosed with PC1/3 deficiency. The variant was inherited uniparental from the carrier mother, who did not have obesity (BMI = 25.4). Further causing doubt into the hypothesis that rare heterozygous *PCSK1* variants can singlehandedly cause obesity is the general lack of reports on obesity from first-degree relatives of patients with PC1/3 deficiency. To date, all PC1/3 deficiency cases carry either homozygous or compound heterozygous *PCSK1* variants, and their carrier relatives only rarely suffer from obesity. Indeed, from the 28 unique PC1/3 deficiency cases, only one carrier parent is reported to be obese (BMI = 31.6) and three carrier parents were reported to be overweight (BMI = 25.4–27.5), as summarized in Table 1. This is also the case for a recent report by Saeed et al. [37], which describes a homozygous nonsynonymous *PCSK1* variant (p.N127I, rs574780528, GRCh38p13 NM_000439.5:c.380A>T) in an obese patient (BMI = 41.5 kg/m^2^, age = 24 years) and his obese sister (BMI = 35 kg/m^2^, age = 30). These individuals are not included in Table 1 due to lack of conclusive phenotypic information regarding clinical PC1/3 deficiency symptoms. The homozygous *PCSK1* variant is suggested to be the cause of their obesity, although none of their three siblings or parents, who carry the same *PCSK1* variant in a heterozygous state, are obese (BMI = 25–27 kg/m^2^). Several different studies with restricted cohort sizes have also looked at the association between rare heterozygous coding *PCSK1* variants and obesity without positive results [35,38,39].

Notwithstanding these reports on absent or restricted association between rare heterozygous *PCSK1* variants and obesity, recent studies with extensive cohorts do make a compelling case for their relevance. In July of 2021, a study in which 645626 exomes from the UK biobank cohort, the MyCode Community Health Initiative cohort and from the Mexico City Prospective Study were included, found an increase of 1.8 BMI units for rare heterozygous nonsynonymous *PCSK1* variants [40]. Interestingly, they found a stronger association with obesity when looking solely at protein truncating variants. A newer study from January of 2022, comprising 199087 exomes from the UK biobank, also investigated the effect of rare genetic variants on BMI [41]. They found that protein-altering *PCSK1* variants significantly increase BMI with a modest 0.34 units. Similarly to the previous study, they found that disruptive *PCSK1* variants (including only nonsense and frameshift variants) have a more pronounced effect (2.29 BMI increase). These results imply a robust and greater effect for scarce protein truncating variants, whereas most nonsynonymous variants have only modest effects with (very) limited clinical relevance. The latter is clearly demonstrated by Curtis et al. [41] by the absence of significant effects on BMI for variants categorized as deleterious, possibly damaging, and probably damaging.

This constitutes a very meaningful finding considering the mere presence of some rare heterozygous *PCSK1* missense variants has been used for diagnosing genetic obesity. Indeed, in a study by Kleinendorst et al., two Dutch individuals received a definitive diagnosis for genetic obesity, solely attributed to the heterozygous *PCSK1* p.Y181H variant [36]. In a recent study by AbouHashem et al. [42], a heterozygous *PCSK1* nonsynonymous variant carrier was also diagnosed with monogenic obesity. As genetic diagnosis of patients can have a profound impact on tailored treatment options and care, we urge that at present, all rare heterozygous *PCSK1* variants should be carefully and individually assessed before genetic obesity diagnosis. A first indication of the possible importance of variants is the type of variant and consequently the degree of protein dysfunction. The former is clearly depicted in the higher odds ratios for rare *PCSK1* protein truncating variants in the meta-analyses [40,41] and in a study combining rare heterozygous variants in the monogenic obesity genes leptin, leptin receptor, *POMC*, and *PCSK1*, where they found a higher BMI for subjects with high-impact heterozygous variants [14]. The added relevance of functional characterization of variants in genes associated with monogenic obesity is very well illustrated in a recent publication by Wade et al. [43]. They showed that in vitro confirmed loss of function rare variants in the melanocortin 4 receptor gene (a gene known to cause monogenic obesity) have a substantial effect on body weight, whereas studies lacking functional characterization mostly find much smaller effects [44]. Specifically for rare heterozygous *PCSK1* variants, the possibility of a dominant negative effect should also be taken into account, as a recent study has identified two rare heterozygous *PCSK1* variant carriers that were affected with severe obesity as well as unspecified chronic diarrhea [14].

Undoubtedly, more research is necessary to further elucidate the full extent to which (rare) heterozygous *PCSK1* variants affect BMI and obesity. Experimental design and sample size will be of paramount importance to detect small to moderate effects and avoid population stratification as well as ascertainment bias. Valuable information will be obtained when the emphasis moves from genetic screening of cases to large (control) cohort screening, combined with functional characterization of variants and exploration of vertical transmission in families.

## 5. Conclusions

In conclusion, we suggest that the *PCSK1* p.Y181H variant is a founder variant which does not contribute to an increased obesity risk. Together with state-of-the art literature, this currently questions the involvement of rare heterozygous nonsynonymous *PCSK1* variants in monogenic obesity and suggests a spectrum of effects on complex obesity depending on the remaining PC1/3 functionality.

## Figures and Tables

**Figure 1 genes-13-01746-f001:**
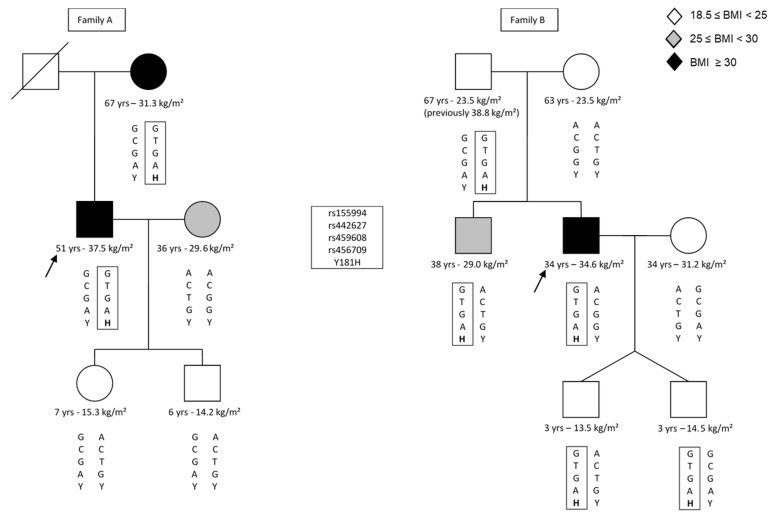
Pedigree of p.Y181H variant carrier families. Pedigrees are shown for both p.Y181H families (family A and B). The probands for each family are marked with an arrow. The first line below the symbols indicates the age of the individual at investigation, followed by his/her BMI. The genotypes for the investigated SNPs in intron 3 (rs155994, rs442627, rs459608, and rs456709, respectively) and p.Y181H in exon 4 are shown, with the common haplotype boxed.

**Table 1 genes-13-01746-t001:** Overview of PC1/3 deficiency cases and their carrier relatives. Cases ID 1–25 have been reviewed by and adapted from Pepin et al. [7]. Abbreviations: *n*/a = not applicable, NA = not assessed, BMI = body mass index.

ID	c.Mutation °	p.Mutation °	Mutation Type °	Obesity	Carriers in Family	Health Information of Carriers in Family	References
1	c.1777G>A/c.620 +4A>C	p.Gly593Arg/NA	Missense/Splice Site	yes (<3 years)	4 heterozygous childrenBREAK(3x c.1777G>A, 1x c.620 +4A>C)	c.1777G>A: “All unaffected”BREAKc.620 +4A>C: NA	[5,6]
2	c.748G>T/c.638_640delCAG	p.Glu250*/p.Ala213del	Nonsense/Nonsense	Yes (<18 month)	Heterozygous parents BREAK(mother: c.748G>T, father: c.638_640delCAG)	both “clinically normal”	[15]
3	c.920C>T	p.Ser307Leu	Missense	yes	Heterozygous parents	“not obese”	[16]
4	c.1024delT/c.-775208_*59002del	*n*/a	Frameshift/Gene Deletion	yes (2 years)	heterozygous parents BREAK(mother: frameshift, father: gene deletion)	“normal weight”	[17]
5	c.1777G>A	p.Gly593Arg	Missense	yes (7.7 years)	heterozygous parents	NA	[18]
6	c.625G>A/c.772C>A	p.Gly209Arg/p.Pro258Thr	Missense	NA	NA	NA	[18]
7	c.1095+1G>T	NA	Splice Site	yes (4.3 years)	heterozygous parents	NA	[18,19]
8	c.1009C>T	PGln337*	Nonsense	no (died 8 month)	heterozygous parents	NA	[18]
9, 10	c.1213C>T	p.Arg405*	Nonsense	yes (3.7 years and 9.3 years)	NA	NA	[18,20]
11,12	c.1_2delATinsTA	p.Met1*	Nonsense	yes (2.9 years and 12.8 years)	NA	NA	[18,20]
13	c.1095+1G>A	*n*/a	Splice Site	no (died 15 month)	heterozygous parents	NA	[18]
14	c.1348_1353del	p.Val450Valfs*1	Frameshift	no (5.5 years)	heterozygous parents, 2 heterozygous siblings	NA	[18]
15	c.1643T>C	p.Phe548Ser	Missense	yes (3.5 years)	NA	NA	[18]
16	c.693C>A	p.Tyr231*	Nonsense	yes (2.4 years)	heterozygous parents, heterozygous sibling	NA	[18]
17	c.1269C>G	p.Asn423Lys	Missense	yes (17 years)	NA	NA	[18]
18	c.1029C>A	p.Tyr343*	Nonsense	NA (polyphagia)	NA	NA	[21]
19, 20, 21	c.927C>G	p.Asn309Lys	Missense	yes (6 years), no (12 month and died 5 month)	Heterozygous parents	“healthy parents”	[22]
22	c.544-2A>G	*n*/a	Splice Site	yes (12 month)	heterozygous parents	“healthy parents”	[23]
23, 24	c.1323C>T	p.Arg438*	Nonsense	yes (2 years and 4 years)	NA	NA	[24]
25	c.679del	p.Val227Leufs*12	Frameshift	yes (9 month)	heterozygous parents	mother BMI 27.5, father BMI 26.1	[25]
26	c.595C>T	p.Arg199*	Nonsense	yes (3 years)	heterozygous parents	mother BMI 31.6 father 23.4	[7]
27	NA	p.Arg391*	Nonsense	NA (“excessive weight gain”)	NA	NA	[8]
28	c.685G>T	p.V229F	Missense	yes (14 years)	NA	“obese mother”	[9]
29, 30	c.500A>C	p.Asp167Ala	Missense	yes (1.5 years and 4.5 years)	heterozygous parents	NA	[10]
31, 32	c.1350_1353del	p.Asp451fs	Frameshift	no (6 years), yes (12 years)	NA	NA	[11]
33	c.238C>T	p.Arg80*	Frameshift	no (11 month, weightgain)	heterozygous mother	mother BMI 25.4	[12]
34	c.1034A>C	p.E345A	Missense	no (14 month, weightgain)	NA	NA	[13]
35	c.286-2A>G	*n*/a	Splice Site	yes	NA	NA	[14]

° If there is only one mutation described for c.Mutation, p.Mutation and Mutation Type, the patient is homozygous for this mutation. * The description of the variants.

**Table 2 genes-13-01746-t002:** Population characteristics. Mean values ± standard deviation are shown. BMI z-score is based on the Flemish Growth Charts [27]. Overweight (%) is calculated as follows: BMI/P50 BMI for age and sex ×100. *n*/a not applicable.

Parameter	Lean Controls	Children/Adolescents with Obesity
N	486	481
Men/women	167/319	211/270
Age (years)	35 ± 7	12 ± 4
BMI (kg/m²)	22.1 ± 1.6	30.9 ± 5.8
BMI z-score	*n*/a	2.59 ± 0.54
Overweight (%)	*n*/a	172 ± 25

## Data Availability

The data presented in this study are available on request from the corresponding author.

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
