# Peer review of "Rare Heterozygous PCSK1 Variants in Human Obesity: The Contribution of the p.Y181H Variant and a Literature Review"

_genes, 2022, doi:10.3390/genes13101746_

Round 1

Reviewer 1 Report

The present paper presents the clinical effect of p.Y181H, a heterozygous PCSK1 variant, which causes partial PC1/3 deficiency, resulting in a significantly increased risk for obesity. The quality of content is high, with a logical flow. In the introduction, the authors explain how this variant affects obesity. However, at the end of the introduction section, the authors mention the study design of the present research and state briefly the conclusion. Therefore, I suggest ending the introduction section with the study aims and moving the study design to the methods section and the conclusion at the end of the article. Also, at the beginning of the discussion section, please add a short paragraph about the current research and main findings, continuing by discussing the results of other studies.

Author Response

We thank the reviewer for the appreciation of our work and manuscript and are confident we were able to address the points raised in the revision of the manuscript.  

Point 1: The present paper presents the clinical effect of p.Y181H, a heterozygous PCSK1 variant, which causes partial PC1/3 deficiency, resulting in a significantly increased risk for obesity. The quality of content is high, with a logical flow. In the introduction, the authors explain how this variant affects obesity. However, at the end of the introduction section, the authors mention the study design of the present research and state briefly the conclusion. Therefore, I suggest ending the introduction section with the study aims and moving the study design to the methods section and the conclusion at the end of the article.

Response 1: The study design and the brief conclusion have been deleted from the introduction, which now ends with the aims of the study. The study design has been described in detail in the methods section and there is a clear conclusion at the end of the article.

Point 2: Also, at the beginning of the discussion section, please add a short paragraph about the current research and main findings, continuing by discussing the results of other studies.

Response 2: Due to contextual preferences, the short paragraph about the current research findings can be found relatively at the beginning of the discussion section, starting at line 218.

Reviewer 2 Report

This manuscript by Dr. Evelien Van Dijck et al investigated the effect of Heterozygous PCSK1 Variants in Human Obesity, and concluded that rare heterozygous PCSK1 variants are rarely associated to monogenic obesity as first-degree carrier relatives of patients with PC1/3 deficiency are mostly not reported to be obese. Although the data are interesting, the results are negative findings.

Author Response

We thank the reviewer for the appreciation of our data.  However, we would not consider our data as “negative findings” in a way that nothing was found, if that is meant by the reviewer. We generated evidence for a founder effect of the most common variant and, partially based on this, were able to conclude that there is only little indication for the causality of heterozygous PCSK1 mutations in obesity.  As this is in contrast with some previous reports, it has important implications in the context of molecular genetic diagnosis and councelling.
